# Evolution of Grain Structure and Dynamic Precipitation during Hot Deformation in a Medium-Strength Al-Zn-Mg-Er-Zr Aluminum Alloy

**DOI:** 10.3390/ma16124404

**Published:** 2023-06-15

**Authors:** Jiongshen Chen, Li Rong, Wu Wei, Peng Qi, Meng Wang, Zezhong Wang, Li Zhou, Hui Huang, Zuoren Nie

**Affiliations:** 1Key Laboratory of Advanced Functional Materials, Education Ministry of China, Beijing University of Technology, Beijing 100124, China; chenjionshen@emails.bjut.edu.cn (J.C.);; 2Xi’an Aerospace Power Machinery Corporation, Xi’an 710025, China; 3Dongfeng Motor Corporation, Wuhan 430056, China

**Keywords:** Al-Zn-Mg-Er-Zr alloy, microstructure evolution, dynamic precipitation, precipitation behavior, hot deformation

## Abstract

The hot deformation behavior of Al-Zn-Mg-Er-Zr alloy was investigated through an isothermal compression experiment at a strain rate ranging from 0.01 to 10 s^−1^ and temperature ranging from 350 to 500 °C. The constitutive equation of thermal deformation characteristics based on strain was established, and the microstructure (including grain, substructure and dynamic precipitation) under different deformation conditions was analyzed. It is shown that the steady-state flow stress can be described using the hyperbolic sinusoidal constitutive equation with a deformation activation energy of 160.03 kJ/mol. Two kinds of second phases exist in the deformed alloy; one is the *η* phase, whose size and quantity changes according to the deformation parameters, and the other is spherical Al_3_(Er, Zr) particles with good thermal stability. Both kinds of particles pin the dislocation. However, with a decrease in strain rate or increase in temperature, *η* phases coarsen and their density decreases, and their dislocation locking ability is weakened. However, the size of Al_3_(Er, Zr) particles does not change with the variation in deformation conditions. So, at higher deformation temperatures, Al_3_(Er, Zr) particles still pin dislocations and thus refine the subgrain and enhance the strength. Compared with the *η* phase, Al_3_(Er, Zr) particles are superior for dislocation locking during hot deformation. A strain rate ranging from 0.1 to 1 s^−1^ and a deformation temperature ranging from 450 to 500 °C form the safest hot working domain in the processing map.

## 1. Introduction

Al-Zn-Mg aluminum alloy is a kind of heat-treatable aluminum alloy that can obtain excellent mechanical properties and corrosion resistance through a proper heat treatment process [1,2,3]. However, with the development of lightweight structural parts, improvement in the processing properties of Al-Zn-Mg alloys is required [4,5]. The low-copper, medium-strength, weldable Al-Zn-Mg alloys, such as 7003, 7005, 7020 and 7N01, have excellent processing properties compared with the high-strength Al-Zn-Mg alloys, and so can produce complex aluminum alloy structural parts through integral forming technology [6,7,8,9]. However, the relatively low strength of the alloy limits its application. If the content of the main alloying elements (Zn, Mg) is increased, the strength of the alloy can be improved significantly, but the processing properties of the alloy will be decreased correspondingly [10,11]. In order to maintain good strength and processing properties at the same time, adding some trace elements, such as V [12], Sc [13,14], Zr [15], Er [16], etc., to modify the alloys is one solution. Babaniaris et al. observed that Al_3_(Sc, Zr) dispersoids can pin dislocations and so improve the strength, but the effect of Al_3_(Sc, Zr) dispersoids on the strength of the alloy was found to be temperature-dependent [13]. In recent years, Nie and his colleagues improved the comprehensive performance of aluminum alloy through the addition of Er and Zr, and found that the formation of a kind of thermally stable Al_3_(Er, Zr) dispersoid can effectively pin the dislocation and subgrain boundaries, thus improving the strength and corrosion resistance of the aluminum alloys [16,17,18]. However, the steady-state plastic deformation processing conditions in medium-strength Al-Zn-Mg-Er-Zr alloys and the dynamic precipitation of Al-Zn-Mg-Er-Zr alloy during hot deformation have not been studied in detail.

During hot deformation, the microstructure is highly correlated with the deformation parameters, so it is very important to establish the relationship between the hot deformation parameters and the microstructure evolution. During the hot deformation process of Al-Zn-Mg alloy, dynamic recovery (DRV), dynamic recrystallization (DRX) and dynamic precipitation (DPN) influence each other and lead to changes in the microstructure [19]. Xu observed the effect of the Z parameter on the dynamic precipitation mechanism of Al-Zn-Mg-Cu alloy in hot deformation, and found that a high lnZ value is conducive to precipitation behavior [14]. Sun et al. observed that the large second-phase particles in 7075 alloy would hinder the slip of subgrain boundaries, while the small second-phase particles would promote continuous dynamic recrystallization through rotation of subgrain [20]. Jiang et al. observed that the degree of initial supersaturation controlled the nucleation rates and strongly influenced the dynamic and static precipitation, and the flow stress curves [21]. In general, it has been proved in recent years that the hot deformation parameters affect the dynamic precipitation mechanism, and at the same time, the precipitation has an effect on the deformation structure, texture and recrystallization of the alloy [8,19,20,21,22,23,24,25]. However, the existence forms of different second phases under different hot deformation parameters and their effects on the hot deformation behavior of alloys has scarcely been studied.

Er-Zr microalloying has gradually become an effective way to improve the comprehensive performance of aluminum alloys. However, there has not been sufficient focus dedicated to analyzing the hot deformation behavior of Al-Zn-Mg-Er-Zr alloys. In this study, the hot deformation behavior of Al-Zn-Mg-Er-Zr alloy at strain rates ranging from 0.01 s^−1^ to 10 s^−1^ and temperatures ranging from 350 to 500 °C was studied through isothermal compression experiments. The hot deformation constitutive model was established, and the steady-state plastic working window was obtained by processing maps. The precipitation of the *η* phase and Al_3_(Er, Zr) phase and its effect on the grain and subgrain microstructure of Al-Zn-Mg-Er-Zr alloy under different hot deformation conditions were analyzed.

## 2. Materials and Experimental Methods

The chemical composition of the experimental Al-Zn-Mg-Er-Zr alloy is shown in Table 1. As-cast alloy was homogenized at 280 °C/12 h + 460 °C/20 h and then cooled to room temperature. The homogenized alloy was machined into a cylindrical specimen with a size of Φ8 × 12 mm. First, the sample was heated to the deformation temperature at the heating rate of 5 °C/s and held at the deformation temperature for 3 min, and then isothermal compression was carried out. The specimens were compressed at the temperatures of 350, 400, 450 and 500 °C, and at the strain rates of 0.01, 0.1, 1 and 10 s^−1^, respectively, and at the true strain rate of 0.60 using the Gleeble-3500 experimental machine. Finally, the compressed specimens were immediately water-quenched to preserve the microstructure formed during thermal compression.

The samples for microscopic characterization were sectioned parallel to the compression axis of the deformed cylindrical specimens. Then, the sections were mechanically polished and electropolished for 15 s in the mixed acid solution of 30% HNO_3_ and 70% CH_3_OH at the voltage of ~15 V DC. The deformation microstructure, grain orientation and recrystallization degree of the alloy was performed using the Gemini SEM 300 FEG scanning electron microscope EBSD probe. Samples for TEM analysis were prepared by mechanically polishing the samples to the thickness of approximately 100 μm and subsequently punching them into 3 mm discs, and finally, twin-jet polishing with a solution of 30% nitric acid and 70% methanol electrolyte at a ~15 V DC voltage and a temperature below −25 °C was performed.

## 3. Results

### 3.1. Initial Microstructure

The microstructures of the Al-Zn-Mg-Er-Zr alloy before deformation are shown in Figure 1. After homogenization, there were still some residual phases in the alloy. According to the EDS analysis results (Table 2) combined with the phase diagram, these residual phases were mainly an Al_8_Cu_4_Er phase and AlZnMg phases containing Er. There were a large number of precipitated phases with a size of about 25 nm in the grain, as shown in Figure 1b. Combined with the inserted diffraction spots in Figure 1b, it can be concluded that these precipitants were Al_3_(Er, Zr) particles with an L1_2_ lattice structure.

### 3.2. Flow Curves

Figure 2 shows the flow stress-strain curve selected in the isothermal compression of Al-Zn-Mg-Er-Zr alloys. At the initial deformation stage, with the strain increases, the dislocation increased significantly, and the flow stress of the alloy increased rapidly due to work hardening. With the further increase in strain, the accumulated energy provided sufficient driving force for the dislocation movement, resulting in dynamic softening mechanisms, such as DRV and DRX. The flow stress of the alloy increased rapidly due to work hardening at the initial deformation stage, and then the flow stress curve of the alloy reached a long-term dynamic equilibrium state due to the combined action of work hardening and dynamic softening. The increase in the deformation temperature provided more driving force for dislocation movement; meanwhile, the decrease in the deformation rate provided enough time for dislocation movement, so the flow stress decreased with the increase in temperature or the decrease in strain rate.

### 3.3. Constitutive Model

Constitutive modeling was used to estimate the thermal deformation behavior of the alloy and construct the relationship between flow stress, deformation temperature T, strain rate and deformation quantity. The Arrhenius-type equation, which includes the Zener-Hollomon parameter, is generally used to describe the hot deformation behavior of Al-Zn-Mg alloy. There are three different functions that can express the dependence of the flow stress on the deformation temperature and strain rate in high-temperature deformation. The power law equation, as shown in Equation (1), is suitable for low-stress deformation, while the exponential law equation, as shown in Equation (2), is suitable for high-stress deformation, and finally, the hyperbolic-sine-type equation, as shown in Equation (3), is suitable for all stress levels [26,27].
(1)$$\dot{\varepsilon }=A_{1}exp\left(\frac{-Q}{RT}\right)\sigma^{n_{1}}\left(\alpha \sigma <0.8\right)$$
(2)$$\dot{\varepsilon }=A_{2}exp\left(\frac{-Q}{RT}\right)exp\left(\beta \sigma \right)\left(\alpha \sigma >1.2\right)$$
(3)$$\dot{\varepsilon }=Aexp\left(\frac{-Q}{RT}\right){\left[\sin\left(\alpha \sigma \right)\right]}^{n}\left(for\;all\;\sigma \right)$$
where *σ* is the flow stress; $\dot{\varepsilon }$ is the strain rate; T is the absolute temperature; *Q* is the deformation activation energy; R is the universal gas constant (8.314 J mol^−1^K^−1^); *A*, *A*_1_, *A*_2_, n, n_1_, *α* and *β* are the calculated material constants. The *Z* parameter, which can be used to express the linear relationship between flow stress and peak stress, is shown as follows [28]:(4)$$Z=\dot{\varepsilon }exp\left(\frac{Q}{RT}\right)=A{\left[\sin h\left(\alpha \sigma \right)\right]}^{n}$$

Taking the natural logarithm on both sides of Equations (1)–(3), the linear relationship between ln⁡σ, *σ*, $\ln\sin\left(\alpha \sigma \right)$ and lnε can be obtained, as shown in Figure 3a–c. The average value of the slopes in Figure 3a, which is the value of n_1_, was 6.78936. The correlation coefficient was 94.63%, as shown in Figure 3a. The average value of the slopes in Figure 3b, which is the value of *β* (MPa^−1^), was 0.1057325, and the correlation coefficient was 92.72%. The parameter α can be defined as *α* = *β*/n_1,_ α = 0.0154260 MPa^−1^. The average value of the slopes in Figure 3c, which is the value of n, was 4.93996, and the correlation coefficient was 94.47%. The average value of the slopes in Figure 3d, which is the value of *Q*/nR, was 3.89649, and the correlation coefficient was 96.00%. The calculated activation energy *Q* was 160.03 kJ·mol^−1^. Deformation activation energy *Q* is a critical physical parameter for measuring the degree of deformation difficulty in hot deformation. Under the same deformation condition, flow stress σ decreased with the decrease in deformation activation energy *Q*, so the alloy with a lower *Q* value was deformed more easily with a lower force, as shown in Equation (3). The self-diffusion activation energy of pure aluminum is 142 kJ·mol^−1^. In the medium-strength weldable Al-Zn-Mg alloys with a low copper content, the deformation activation energy *Q* of the studied alloy was higher than that of the 7N01 alloy (152 kJ·mol^−1^), as reported by Liu et al. [29], but lower than that of the 7039 (172.7 kJ·mol^−1^), as alloy reported by Li et al. [30] Compared with high-strength Al-Zn-Mg alloys, the deformation activation energy *Q* of the studied alloy was lower than that of the 7075 (269.04 kJ·mol^−1^) [31] alloy and 7150 (229.75 kJ·mol^−1^) alloy [32]. So, the formability of the studied alloy was relatively good among the Al-Zn-Mg alloys.

Taking the logarithm of both sides of Equation (4), the following equation can be obtained:(5)$$\ln Z=\ln A+n\ln\sin h\left(\alpha \sigma \right)$$

The relationship between ln *Z* and $\ln\sin h\left(\alpha \sigma \right)$ could be obtained through linear fitting, and the correlation was 95.87%, as shown in Figure 4. The *Z* parameter decreased with the increase in deformation temperature and the decrease in strain rate.

The value of ln *A* and n is the intercept and the slope in the $\ln\sin h\left(\alpha \sigma \right)-ln\;Z$ plot, respectively, and so the value of ln *A* and n could be obtained as 25.53028 and 4.79425. Thus, the relationship between flow stress and the *Z* parameter is as follows:(6)$$\sigma =58.90\ln\left\{\left.{\left(\frac{Z}{1.223657E^{11}}\right)}^{0.21}+{\left[{\left(\frac{Z}{1.223657E^{11}}\right)}^{0.42}+1\right]}^{0.5}\right\}\right.$$

### 3.4. Processing Maps

Prassad [33,34] proposed the dynamic material model (DMM), believing that the external energy from thermal processing is conferred on the alloy, some of which is used for plastic deformation (*G*) and some for structural deformation (*J*), as follows:(7)$$P=\sigma \dot{\varepsilon }=G+J=\int_{0}^{\dot{\varepsilon }}\sigma d\dot{\varepsilon }+\int_{0}^{\sigma }\dot{\varepsilon }d\sigma$$
where the ratio of *G* to *J* is determined by the strain rate sensitivity factor *m*:(8)$$m=\frac{dJ}{dG}=\frac{\dot{\varepsilon }d\sigma }{\sigma d\dot{\varepsilon }}$$

The power dissipative efficiency and instability coefficient are defined by the matrix that formed according to the deformation temperature T and strain rate *η*:(9)$$\eta =\frac{2m}{m+1}$$
(10)$$\xi \left(\dot{\varepsilon }\right)=\frac{\partial \ln\left(\eta \right)}{\partial \ln\dot{\varepsilon }}+m<0$$

The power dissipation efficiency represents the ratio of the energy consumed by the microstructure change to the energy consumed by the ideal linear dissipation state. Generally, the higher the power dissipative efficiency is, the better the deformation performance will be. In addition, when the instability coefficient $\xi \left(\dot{\varepsilon }\right)<0$, the alloy will suffer from rheological instability, and the plasticity and toughness of the alloy will decrease significantly, resulting in crushing and cracking. Therefore, the higher the power dissipation efficiency of the alloy and outside the instability zone, the better the material forming performance [35].

Figure 5 shows the 3D efficiency maps of the experimental alloy under different strains. Warm color represents high power dissipation efficiency and good plastic working properties. It can be seen that in the strain rate between 0.1 s^−1^ and 10 s^−1^, the power dissipative efficiency increased with the increase in deformation temperature. This is because high temperatures enhance the effect of thermal activation and promote microstructure evolution such as DRV and DRX during the hot deformation of the studied alloy. When the strain was 0.1, the experimental alloy had three high dissipation efficiency regions, as shown in Figure 5a: (1) 0.05–1 s^−1^ and 450–500 °C; (2) 0.01–0.05 s^−1^ and 430–480 °C; (3) 0.01–0.03 s^−1^ and 350–375 °C. The dissipation efficiency of these three regions exceeded 30%. With the increase in strain, the DOM#2 region gradually decreased and then disappeared when the strain reached 0.3. The DOM#3 region also gradually decreased until it disappeared when the strain reached 0.5. However, the temperature range suitable for processing in the DOM#1 region expanded with the increase in strain. In general, with the increase in strain, the area of high dissipation efficiency of the investigated Al-Zn-Mg-Er-Zr alloy gradually decreased. When the strain reached 0.6, only DOM#1 had high dissipation efficiency (0.1~1 s^−1^ and 425~500 °C), as shown in Figure 5. The power dissipation efficiency increased with the increase in strain in DOM#1. This is because as the strain increased, the stored energy increased as the dislocation accumulated.

By superimposing the processing instability diagram onto the power dissipation efficiency diagram of the experimental alloy, we can obtain the thermal processing diagram of the experimental alloy, as shown in Figure 6. The gray area is the instability area of plastic workability. It can be observed that the material instability region appeared at a high strain rate and low deformation temperature. The main reason for this is that grain boundary sliding is difficult at a low temperature, which leads to stress concentration at grain boundaries; when the strain rate is high, the deformation time interval is so small that it is not sufficient for dynamic softening, and so stress concentration at grain boundaries could have been the source of cracking in the experimental alloy during hot working. Thus, the suitable processing region of the experimental alloy was determined to be a deformation temperature between 450 °C and 500 °C and strain rate between 0.1 s^−1^ and 1 s^−1^.

## 4. Discussion

### 4.1. Grain and Subgrain Microstructures

In order to explore the microstructure evolution of experimental alloy during thermal processing, microstructure analysis and discussion were carried out on the deformation samples under different processing conditions in the thermal processing diagram when the deformation was 0.6. It can be seen from Figure 7 that after hot deformation, the grains were elongated and extended in the direction perpendicular to the compression axes. When the deformation temperature was 350 °C, as shown in Figure 7a, the alloy was in the processing instability region, and the wrought structure dominated in the alloy with a proportion of about 98.73%, as shown in Figure 7e. When the temperature increased to 400 °C, as shown in Figure 7f, the microstructure showed obvious dynamic recovery; the reason for this phenomenon is that a higher deformation temperature provides a higher driving force for dislocation motion and the formation of polygonal subgrain boundaries. After that, with the increase in temperature, the proportion of wrought structure in the experimental alloy decreased, and the proportion of recrystallization structure increased. In general, in the stability domain, as shown in Figure 7, when the deformation temperature increased, sufficient driving force was provided for the experimental alloy, dislocation density decreased, the subgrain structure was dominant, and the main softening mechanism of the alloy was dynamic recovery.

Figure 8 shows the EBSD results of the alloy at different strain rates. When the strain rate was 0.01 s^−1^, the deformed structure in the alloy accounted for 18.93%, so the power dissipation efficiency was low at 18.3%. When the strain rate was 0.1 s^−1^, the proportion of deformed structure of the alloy gradually decreased to 10.84%, the dynamic recovery of the alloy was relatively complete, and the power dissipation efficiency was 40.8%. When the strain rate was increased to 1 s^−1^, a large number of recrystallized grains of a smaller size appeared at the subgrain boundary, but the proportion of deformed structure increased to 19.88%, so the power dissipation efficiency decreased to 35.6%. When the strain rate rose to 10 s^−1^, because the deformation time was too short, the dislocation did not decrease or increase quickly enough to cut elongated deformation grains to form polymorphic subgrain boundaries. So, a large number of deformed structures were preserved inside the alloy, accounting for 47.01%, and the power dissipation efficiency was 3.0% at this time. During the thermal deformation of the alloy, the energy provided by the external environment for the deformation of the alloy structure was partly used for the dynamic recovery and recrystallization, and partly for the nucleation and growth of the precipitated phase in the alloy. When the strain rate was low, the alloy had enough deformation time to complete the dynamic recovery, but at the same time, the precipitate had enough time for nucleation growth to hinder the dynamic recovery and recrystallization of the alloy, so the power dissipation efficiency of the experimental alloy was low at the low strain rate. When the strain rate was too high, the deformation time of the alloy was too short, leading to a short dislocation migration time, and thus the dynamic recovery of the alloy was not complete. In this case, the power dissipation efficiency was also very low, so the alloy remained in the processing instability region. Therefore, when the strain rate was too high or too low, it was not suitable for thermal processing. In summary, the experimental alloy thermal processing range is between a deformation temperature of 450 and 500 °C and a strain rate of 0.1~1 s^−1^.

### 4.2. Precipitation Behavior

As shown in Figure 9, the experimental alloy contained two kinds of precipitates. One was the *η* phase with a size between 50 and 200 nm, which could easily change with the change in deformation parameters. The other was spherical Al_3_(Er, Zr) particles with sizes ranging from 10 nm to 20 nm, which had good thermal stability. During thermal compression, the evolution of the microstructure of these particles played an important role. The microstructures of samples deformed at 450 °C and 0.1 s^−1^ are shown in Figure 9, and under these deformation conditions, it reached the stability domains of pressing maps (with a large power dissipation efficiency and significant distance from the instability zone). As can be seen from Figure 9a, there were a large number of *η* phases, and the dislocation wall formed in the subgrain boundaries. In the dislocation wall, as shown in Figure 9b, the *η* phase with a size of 80~150 nm and the Al_3_(Er, Zr) particles with a size of 10~20 nm both played a role in pinning the dislocation movement, and there were still large areas of wrought structure in the interior of the subgrain, which is consistent with the results of EBSD, as shown in Figure 7 and Figure 8. As shown in Figure 9c, there were a large number of dispersed particles with sizes ranging from 10 nm to 20 nm in the grain interior, and the distribution of precipitates was relatively irregular. Superlattice structure spots in the corresponding SAED patterns inserted in Figure 9c confirmed the existence of the Al_3_(Er, Zr) particles. The number density of Al_3_(Er, Zr) particles in the deformed alloys was obviously higher than that in the undeformed alloys, indicating that the dynamic precipitation of Al_3_(Er, Zr) particles also occurs during hot deformation.

TEM images of the Al-Zn-Mg-Er-Zr alloy deformed under different conditions are shown in Figure 10. It is obvious that the deformation temperature and strain rate had great influence on the dynamic precipitation and the evolution of grain structure during hot deformation. Deformed grains containing a high density of tangled dislocations were found in specimens deformed at 350 °C and 10 s^−1^, as shown in Figure 10a. When the temperature rose to 400 °C (Figure 10b), the higher driving force led to greater dislocation motion, and so the entanglement phenomenon disappeared. At 400 °C, the Al_3_(Er, Zr) particles were distributed more uniformly, and the microstructure showed characteristics typical of dynamic recovery. When the deformation temperature rose, the *η* phase became scattered and its size increased, and the subgrain boundary became more distinct, as shown in Figure 10b,c. The strain rate also affected the dynamic precipitation and grain structure. With the decrease in the strain rate, the alloy underwent a longer deformation time. So, the *η* phase dynamically coarsened, and its density decreased. The dislocation locking of particles was weakened, and the dislocation walls gradually transformed into subgrain boundaries. When the temperature rose to 500 °C, the *η* phase did not precipitate any longer.

Al_3_(Er, Zr) particles were distributed in two ways: one is random distribution, and the other is chain arrangement. The random distribution of Al_3_(Er, Zr) particles was dominant at low deformation temperatures, as shown in Figure 10b,c, but when the deformation temperature reached 500 °C, some Al_3_(Er, Zr) particles began to form chains. With the increase in strain rate, the number density of Al_3_(Er, Zr) particles increased, and the chain arrangement became more dominant. This is perhaps because the driving force of precipitation of Er and Zr in α-Al increases exponentially with the increase in the diffusion rate, and these dislocations or substructures provide higher diffusion rates for Er and Zr, and so Al_3_(Er, Zr) nucleates predominately on the dislocations or substructures. Thus, Al_3_(Er, Zr) particles exhibit chain arrangement. It has also been mentioned in previous studies [36,37,38,39] that a large number of dislocations introduced by deformation could accelerate the nucleation, growth and coarsening of precipitated phases. When the strain rate is very high, only the precipitates precipitated dynamically along the dislocation line have enough time to nucleate and grow to be visible under TEM, while those precipitates not precipitated along dislocations have almost no time to grow to a sufficient size. This is consistent with the Al_3_(Er, Zr) precipitation distribution observed in Figure 10e,f. The higher the strain rate, the more Al_3_(Er, Zr)-precipitation-induced chain arrangements.

When the deformation temperature was below 450 °C, the *η* phase also had the effect of pinning dislocation. However, when the temperature was higher than 500 °C, a large number of *η* phases dissolved or coarsened, and their density decreased, and so dislocation locking of the *η* phase reduced significantly. The diffusivity and the solid solubility of Er and Zr in an aluminum matrix are low, and the growth and coarsening of Al_3_(Er, Zr) particles are rare, which is correlated with the thermal stability of precipitated phases. Therefore, the deformation parameters had little effect on the size of Al_3_(Er, Zr) particles; the dislocation locking effect of Al_3_(Er, Zr) particles existed under various deformation conditions and was not reduced with the rise in temperature. So, at higher deformation temperatures, Al_3_(Er, Zr) particles still pin dislocations and thus refine subgrain and enhance the strength. Conclusively, compared with the *η* phase, Al_3_(Er, Zr) particles are superior for dislocation locking.

## 5. Conclusions

In the current work, the grain microstructure and precipitate evolution in Al-Zn-Mg-Er-Zr alloy during hot deformation was investigated. Based on the experimental data, the constitutive equation and the thermal processing maps were established, and the effects of hot deformation conditions on the microstructure (including grain, subgrain and precipitation) of the alloy were analyzed. The microstructure characteristics of the alloys in the steady deformation zone and unstable deformation zone in the thermal processing maps were also analyzed. The conclusions are as follows:(1)The steady-state flow stress increases with the increase in strain rate or the decrease in deformation temperature, and can be described with the hyperbolic sinusoidal constitutive equation with a deformation activation energy of 160.03 kJ/mol.(2)According to the processing maps and EBSD results, when the deformation temperature was 350 °C, the alloy was in the processing instability domain, and the wrought structure dominated in the studied alloy. With the increase in deformation temperature, sufficient driving force was provided for the studied alloy, dislocation density decreased, the subgrain structure was dominant, and the main softening mechanism of the alloy was dynamic recovery. When the strain rate was too high, the deformation time interval of the alloy was too short, leading to a short dislocation migration time, and thus the dynamic recovery of the alloy was not complete. In this case, the power dissipation efficiency was also very low, so the alloy processing was in the processing instability domain. Therefore, a strain rate that is too high or too low is not suitable for hot deformation. In summary, a strain rate ranging from 0.1 to 1 s^−1^ and a deformation temperature ranging from 450 to 500 °C form the safest hot working domain. The microstructural characteristics of the alloy in the safe working domain were predominantly dynamic recovery, accompanied by a small number of recrystallized microstructures.(3)The investigated alloy contained two kinds of second-phase particles during hot deformation. One was the *η* phase, whose size and density changed significantly with the variation in deformation conditions, and the other was the spherical Al_3_(Er, Zr) particles, which had good thermal stability. Both kinds of particles played a role in dislocation locking. When the strain rate was decreased or the deformation temperature was increased, the *η* phase coarsened dynamically and its density decreased, and so its dislocation locking ability was weakened.(4)Al_3_(Er, Zr) particles are distributed in two ways: one is random distribution, and the other is chain arrangement. The random distribution of Al_3_(Er, Zr) particles was dominant at low deformation temperatures, but when the deformation temperature reached 500 °C, some Al_3_(Er, Zr) particles began to show chain arrangement. When the deformation temperature was increased, the number density of Al_3_(Er, Zr) particles increased, and the chain arrangement structure became more dominant. However, the deformation conditions had little effect on the size of Al_3_(Er, Zr) particles. So, compared with the *η* phase, Al_3_(Er, Zr) particles are superior for dislocation locking during hot deformation.

## Figures and Tables

**Figure 1 materials-16-04404-f001:**
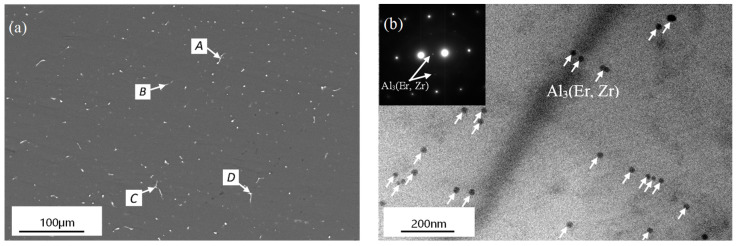
(**a**) SEM and (**b**) TEM images of initial homogenized microstructure.

**Figure 2 materials-16-04404-f002:**
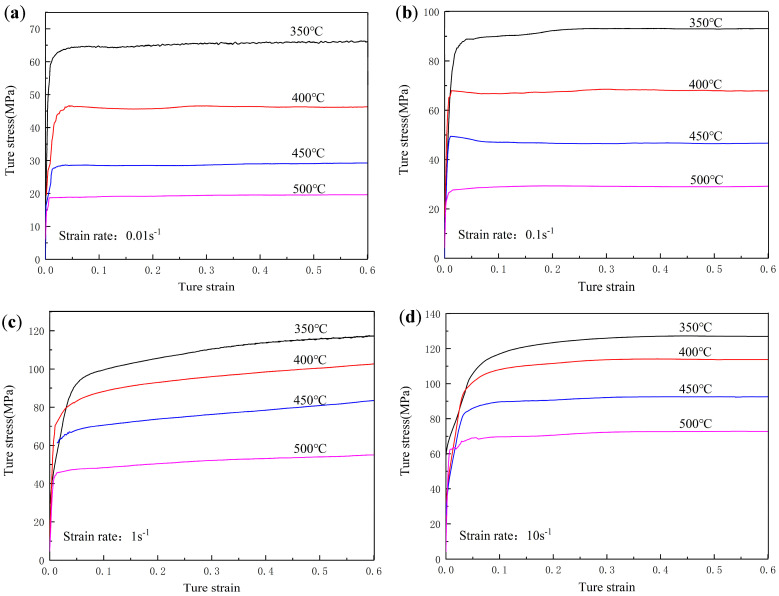
Corrected true stress–strain curves of Al-Zn-Mg-Er-Zr alloy at different strain rates: (**a**) 0.01 s^−1^; (**b**) 0.1 s^−1^; (**c**) 1 s^−1^; (**d**) 10 s^−1^.

**Figure 3 materials-16-04404-f003:**
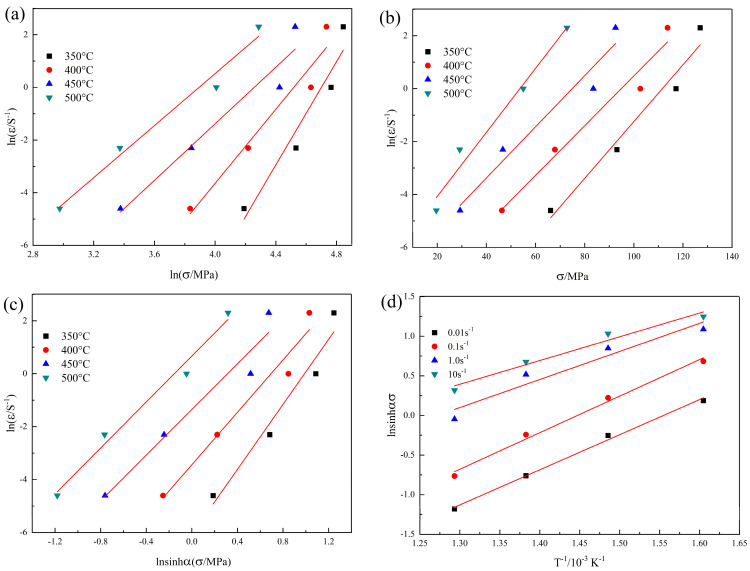
Fitting curves describing the relationship: (**a**) $\ln\dot{\varepsilon }$ − ln⁡σ; (**b**) $\ln\dot{\varepsilon }$ − σ; (**c**) $\ln\dot{\varepsilon }$ − $\ln\sin\left(\alpha \sigma \right)$; (**d**) $\mathrm{lnsinh}(\alpha \sigma )-T^{-1}$.

**Figure 4 materials-16-04404-f004:**
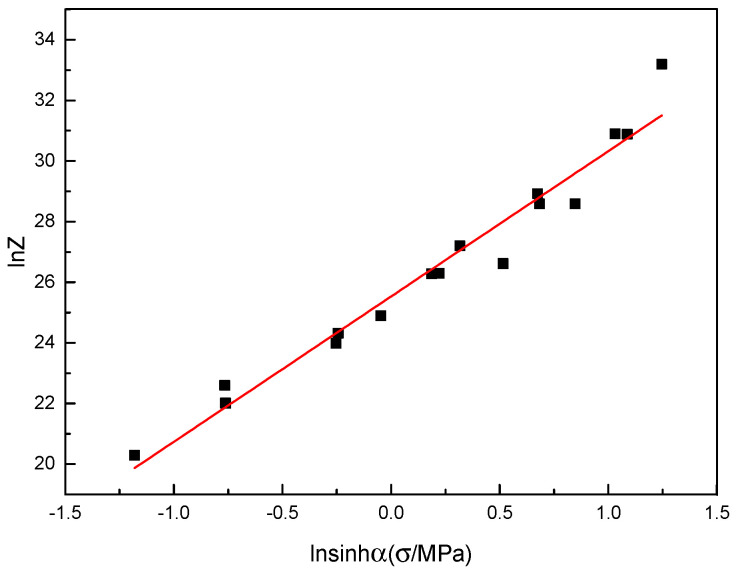
Linear fitting relationship between ln *Z* and $\ln\sin h\left(\alpha \sigma \right)$.

**Figure 5 materials-16-04404-f005:**
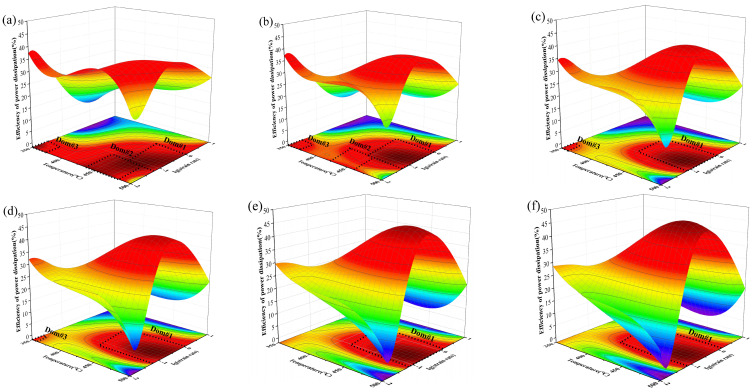
Three-dimensional diagram of the power dissipation efficiency of Al-Zn-Mg-Er-Zr alloy at various strains: (**a**) 0.1; (**b**) 0.2; (**c**) 0.3; (**d**) 0.4; (**e**) 0.5; (**f**) 0.6.

**Figure 6 materials-16-04404-f006:**
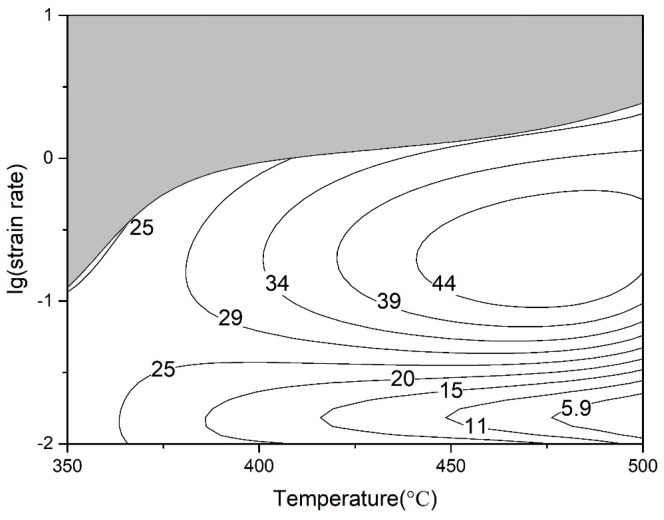
Processing maps of the alloy at the true strain of 0.6.

**Figure 7 materials-16-04404-f007:**
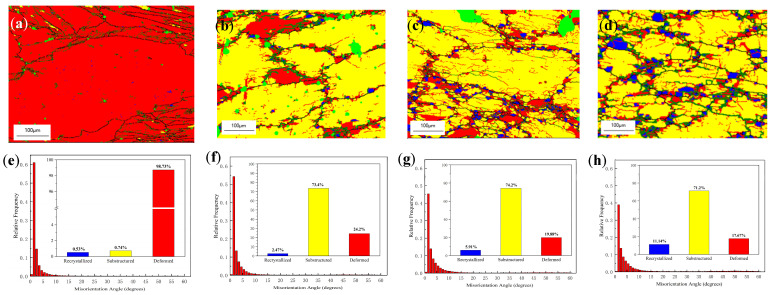
EBSD results of alloy deformation at (**a**,**e**) 350 °C, 1 s^−1^; (**b**,**f**) 400 °C, 1 s^−1^; (**c**,**g**) 450 °C, 1 s^−1^; (**d**,**h**) 500 °C, 1 s^−1^. (**a**–**d**) the misorientation angle distribution maps; (**e**–**h**) histogram of the misorientation angle distributions. (Red: deformed structure; yellow: sub-grains; blue: recrystallized grains).

**Figure 8 materials-16-04404-f008:**
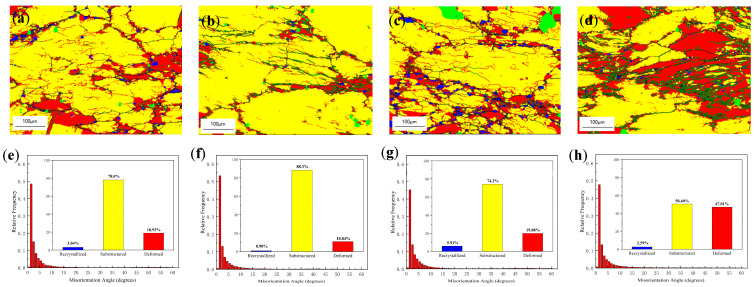
EBSD results of the alloy deformed at (**a**,**e**) 450 °C, 0.01 s^−1^; (**b**,**f**) 450 °C, 0.1 s^−1^; (**c**,**g**) 450 °C, 1 s^−1^; (**d**,**h**) 450 °C, 10 s^−1^. (**a**–**d**) the misorientation angle distribution maps; (**e**–**h**) histogram of the misorientation angle distributions. (Red: deformed structure; yellow: sub-grains; blue: recrystallized grains).

**Figure 9 materials-16-04404-f009:**
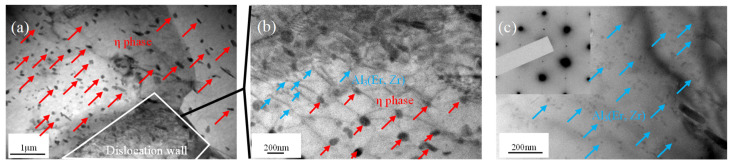
TEM images of Al-Zn-Mg-Er-Zr alloy after deformation at 450 °C and 0.1 s^−1.^ (**a**) Grain boundary; (**b**) Dislocation wall; (**c**) Internal grain. (Red: *η* phase; blue: Al_3_(Er, Zr) particles).

**Figure 10 materials-16-04404-f010:**
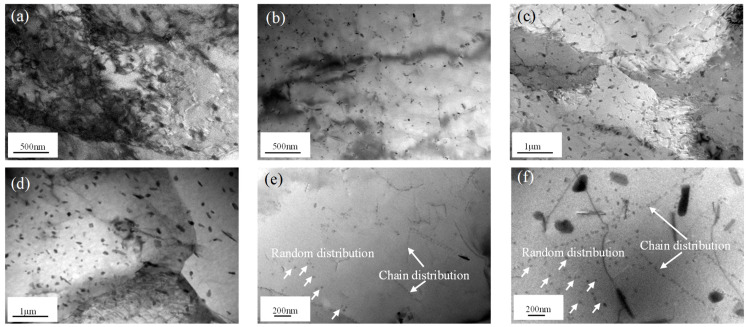
TEM images showing the typical morphologies of precipitate in the Al-Zn-Mg-Er-Zr alloys deformed at: (**a**) 350 °C, 10 s^−1^; (**b**) 400 °C, 10 s^−1^; (**c**) 450 °C, 10 s^−1^; (**d**) 450 °C, 0.1 s^−1^; (**e**) 500 °C, 1 s^−1^; (**f**) 500 °C, 10 s^−1^.

**Table 1 materials-16-04404-t001:** Chemical composition of the experimental alloy (wt%).

Element	Zn	Mg	Cu	Mn	Er	Zr	Si	Al
Concentration	4.53	2.58	0.50	0.54	0.23	0.16	0.04	Bal

**Table 2 materials-16-04404-t002:** EDX results of the residual particles in Al-Zn-Mg-Er-Zr alloy after homogenization (At%).

Zone	Al	Zn	Mg	Cu	Mn	Er
A	82.65	4.72	4.02	5.89	1.02	1.70
B	86.71	2.79	2.52	-	-	7.99
C	81.89	4.43	2.89	8.25	-	2.55
D	84.47	3.46	3.90	6.22	-	1.94

## Data Availability

The data that support the findings of this study are available from the corresponding author, Wu Wei, upon reasonable request.

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
