# Peer review of "Evolution of Grain Structure and Dynamic Precipitation during Hot Deformation in a Medium-Strength Al-Zn-Mg-Er-Zr Aluminum Alloy"

_materials, 2023, doi:10.3390/ma16124404_

Round 1
Reviewer 1 Report
Manuscript ID: materials-2415442
Title: Evolution of grain structure and dynamic precipitation during hot deformation in a medium strength Al-Zn-Mg-Er-Zr aluminum alloy
Overall, the manuscript is well-organized and well-written, but the scientific content is not interesting due to a lack of novelty. The present work and idea are NOT novel. There are many studies in the literature made by the same research group as well as by other scientists. No new findings or conclusions were reported in comparison to previous studies. Accordingly, it is difficult, from my point of view, to recommend the publication of the present study in this reputed journal.
A few typo-mistakes and grammatical errors should be corrected.
Author Response
Dear Reviewer:
Thanks for your comment concerning our manuscript entitled “Evolution of grain structure and dynamic precipitation during hot deformation in a medium strength Al-Zn-Mg-Er-Zr aluminum alloy” (materials-2415442). Your comment is all valuable and very helpful for revising and improving our paper, as well as the important guiding significance to our researches.
Er-Zr microalloying has gradually become an effective way to improve the comprehensive performance of aluminum alloys. But the hot deformation behavior of the heat-treatable Al-Zn-Mg-Er-Zr alloys has not been sufficiently studied. And the existence forms of different second phases under different hot deformation parameters and their effects on the hot deformation behavior of the alloys is scarcely studied. Al3(Er, Zr) particles in Al-Er-Zr alloy and η phase in Al-Zn-Mg alloy precipitated during hot deformation, which will affect the grain and subgrain microstructures of the alloy, have been reported. However, due to the dynamic precipitation of η phase and Al3(Er, Zr) particles in Al-Zn-Mg-Er-Zr alloy, the interaction between η phase and Al3(Er, Zr) particles and their joint influence on the hot deformation behavior will be more complicated.
So in this paper, the hot deformation behavior of Al-Zn-Mg-Er-Zr alloy at strain rates ranging from 0.01s−1 to 10s−1 and temperatures ranging from 350 to 500℃ has been studied by isothermal compression experiments. The hot deformation constitutive model was established and the steady-state plastic working window was obtained by processing maps. The precipitation of η phase and Al3(Er, Zr) phase and its effect on the grain and subgrain microstructure of Al-Zn-Mg-Er-Zr alloy under different hot deformation conditions were analyzed.
We have studied your comment carefully and made corrections and additions, hoping for your approval.

Reviewer 2 Report
The authors make a study of the hot deformation of a medium strength Al-Zn-Mg-Er-Zr aluminum alloy. It is an interesting job, however some questions must be clear:
1. Figure 7 from e to h has to improve, same for 8.
2. Make a clearer discussion of the Constitutive model and the experimental part.
3. What is the reason that the temperature increases and not the particle size?
4. What would be the possible applications?
Author Response
Dear Reviewer:
Thanks for your comments concerning our manuscript entitled “Evolution of grain structure and dynamic precipitation during hot deformation in a medium strength Al-Zn-Mg-Er-Zr aluminum alloy” (materials-2415442). Those comments are all valuable and very helpful for revising and improving our paper, as well as the important guiding significance to our researches. We have studied comments carefully and have made a correction which we hope meet with approval. Revised portions are marked in red in the paper. The main corrections in the paper and the responses to the reviewer’s comments are as following:
- Figure 7 from e to h has to improve, same for 8.
Reply: Thank you for your valuable comments. According to your opinion, we have reprocessed the pictures in the text, and image clarity has been optimized.
- Make a clearer discussion of the Constitutive model and the experimental part.
Reply: Thank you for your valuable comments. We add comparison and discussion in the Constitutive model and the experimental part. (Lines 165-176)
- What is the reason that the temperature increases and not the particle size?
Reply: â‘ The addition of Er and Zr can form thermodynamically stable Al3M phase with aluminum matrix.
â‘¡ The diffusivity of Er and Zr in aluminum matrix is low, the growth of precipitated phase is related to coarsening process and diffusion, and the low diffusivity is beneficial to the thermal stability of precipitated phase.
â‘¢ The solid solubility of Er and Zr in aluminum matrix is low, and the precipitated phase does not solve at a higher temperature and remains stable.
Thank you for your valuable comments. We add related discussion in the paper. (Lines 355-337)
- What would be the possible applications?
Reply: Microalloying has gradually become an effective way to improve the comprehensive performance of aluminum alloys. Sc element used for microalloying in scientific research is expensive and difficult to be widely used in industrial fields. The price of Er element is only 1/80~1/100 the price of Sc, and can also play a role in improving alloy properties, and has been applied in aerospace, automobile manufacturing and other fields.

Reviewer 3 Report
The work contains an applied research. I do not consider that there should be changes in the research methodology, I declare myself satisfied.
Please develop the conclusions further. The conclusions are too short.
Please check the Template, there are some problems related to the Template.
I think that figure 2 is insufficiently explained !
I believe that figure 4 is insufficiently explained !
Pay attention to the title of chapter 4
Figure 5 is not sufficiently explained, it needs an explanation, "why", "how", which is the reason that leads to the explanations provided by the authors between lines 166 and 177. Please make some corrections related to how the research methodology is implemented in your research.
Regarding the bibliography, It is ok.
I cannot make a suggestion related to a certain specific part of the work, but I ask the authors to check the English language of the entire work.
Author Response
Dear Reviewer:
Thanks for your comments concerning our manuscript entitled “Evolution of grain structure and dynamic precipitation during hot deformation in a medium strength Al-Zn-Mg-Er-Zr aluminum alloy” (materials-2415442). Those comments are all valuable and very helpful for revising and improving our paper, as well as the important guiding significance to our researches. We have studied comments carefully and have made a correction which we hope meet with approval. Revised portions are marked in red in the paper. The main corrections in the paper and the responses to the reviewer’s comments are as following:
- Please develop the conclusions further. The conclusions are too short.
Reply: Thank you for your valuable comments. Conclusions on the evolution of grain structure with deformation temperature and strain rate are indeed lacking. We add the conclusions about it. (Lines 375-384)
- Please check the Template, there are some problems related to the Template.
Reply: Thank you for your valuable comments. We download the proper one for revision.
- I think that figure 2 is insufficiently explained !
Reply: Thank you for your valuable comments. We add related discussion in the paper. (Lines 125-132,135-137)
- I believe that figure 4 is insufficiently explained !
Reply: Thank you for your valuable comments. We add related discussion in the paper. (Lines 181-182)
- Pay attention to the title of chapter 4
Reply: Errors in chapter 4 have been corrected. (Line 240)
- Figure 5 is not sufficiently explained, it needs an explanation, "why", "how", which is the reason that leads to the explanations provided by the authors between lines 166 and 177. Please make some corrections related to how the research methodology is implemented in your research.
Reply: Thank you for your valuable comments. The power dissipative efficiency is defined by the matrix which formed by deformation temperature T and strain rate η. The power dissipative efficiency represents the ratio between the energy consumed during the microstructure change and the energy consumed in the ideal linear dissipative state. These are described in the paper. (lines 194-205) In addition, we add related discussion about Figure 5 in the revision. (lines 208-211,221-223)

Reviewer 4 Report
This study investigates the microstructure and precipitate evolution in a new type Al-Zn-Mg-Er-Zr alloy during hot deformation through experimental data, the constitutive equation, and the thermal processing maps. The article is well-organized and written, and the images are high quality. The article's title is practical and attractive, but the following points should be considered before publishing.
The abstract should be written better and needs minor revisions. The purpose of research and innovation should be clearly stated. Also, the performed tests should be presented first, and then the results should be presented quantitatively and qualitatively. For example, the article lacks quantitative data and results. The number of keywords can be increased. Also, the article needs general writing and grammar editing.
The introduction is written very briefly, and at the end, a suitable summary of the importance of the present issue is not provided. The introduction needs to be reformed and deepened. Use the following resources to deepen the introduction. Developing a Mg alloy with ultrahigh room temperature ductility via grain boundary segregation and activation of non-basal slips. Gradient Structure of Ti-55531 with Nano-ultrafine Grains Fabricated by Simulation and Suction Casting. Effect of heat input on interfacial characterization of the butter joint of hot-rolling CP-Ti/Q235 bimetallic sheets by Laser + CMT.
Why are two different heat treatment conditions used to homogenize the microstructure of the prototype? Clarify in this regard. Does the machining operation affect the microstructure of the prototype? Wouldn't it be better to homogenize the microstructure after machining? 350 to 500 degrees Celsius are deformation temperatures? How many temperatures have been selected in this interval? In the research method section, these issues should be clearly stated. How are the reproducibility of compression tests checked? How accurate was the strain and strain rate measurement? In addition to the strain rate, the displacement rate of the device should be mentioned.
Why has the difference in the strength of the sample decreased with the increase in the strain rate? As the strain rate increases, the most significant increase in strength is for the sample with the highest temperature. What is the reason? Why are these results not compared? This weakness can be seen in all parts of the article, including the abstract, results, discussion, and conclusion, which should be solved. How has the used model been verified?
The results section is well organized and categorized. But some parts of it are reporting, and the discussion needs to be deepened. It is suggested to use the following sources. Co-precipitated Ni/Mn shell coated nano Cu-rich core structure: A phase-field study. Phase transformations in an ultralight BCC Mg alloy during anisothermal ageing. From classical thermodynamics to phase-field method. In the discussion section, only two sources have been used.
No comment.
Author Response
Dear Reviewer:
Thanks for your comments concerning our manuscript entitled “Evolution of grain structure and dynamic precipitation during hot deformation in a medium strength Al-Zn-Mg-Er-Zr aluminum alloy” (materials-2415442). Those comments are all valuable and very helpful for revising and improving our paper, as well as the important guiding significance to our researches. We have studied comments carefully and have made a correction which we hope meet with approval. Revised portions are marked in red in the paper. The main corrections in the paper and the responses to the reviewer’s comments are as following:
- The abstract should be written better and needs minor revisions. The purpose of research and innovation should be clearly stated. Also, the performed tests should be presented first, and then the results should be presented quantitatively and qualitatively. For example, the article lacks quantitative data and results. The number of keywords can be increased. Also, the article needs general writing and grammar editing.
Reply: Thank you for your valuable comments. According to your opinion, we have re-edited abstract. (Lines 12-20) For the lack of quantitative data and results in this paper, we added the deformation activation energy Q, which measures the deformation difficulty of the alloy, and made a comparative analysis with other studies. (Lines 165-176)
- The introduction is written very briefly, and at the end, a suitable summary of the importance of the present issue is not provided. The introduction needs to be reformed and deepened. Use the following resources to deepen the introduction. Developing a Mg alloy with ultrahigh room temperature ductility via grain boundary segregation and activation of non-basal slips. Gradient Structure of Ti-55531 with Nano-ultrafine Grains Fabricated by Simulation and Suction Casting. Effect of heat input on interfacial characterization of the butter joint of hot-rolling CP-Ti/Q235 bimetallic sheets by Laser + CMT.
Reply: Thank you for your valuable comments. We add a summary of the importance of the present issue. (Lines 76-78) Thank you for the resources, they are very helpful to us and we used them to deepen the introduction. ([24-26])
- Why are two different heat treatment conditions used to homogenize the microstructure of the prototype? Clarify in this regard. Does the machining operation affect the microstructure of the prototype? Wouldn't it be better to homogenize the microstructure after machining? 350 to 500 degrees Celsius are deformation temperatures? How many temperatures have been selected in this interval? In the research method section, these issues should be clearly stated. How are the reproducibility of compression tests checked? How accurate was the strain and strain rate measurement? In addition to the strain rate, the displacement rate of the device should be mentioned.
Reply: ①Thank you for your valuable comments. In our previous studies, we have optimized the optimal homogenization treatment condition (280℃/12 h + 460℃/20 h) of the alloy, which can both improve the precipitation of Al3(Er, Zr) and dissolve most of the non-equilibrium phase.
â‘¡ We machined the sample by WEDM. And We have made comparative observation before, and have found the processing will not affect the initial microstructure.
③ We selected four deformation temperatures between 350-500℃, i.e. 350℃, 400℃, 450℃ and 500℃ respectively, which have been explained in the experiment part. (Lines 94-97)
â‘£ The Gleeble-3500 testing machine applies force on the sample through the loading system driven by the electro-hydraulic servo valve. In this experiment, the strain and strain rate are set by controlling the stroke and moving time of the hydraulic cylinder piston (the stroke is accurate to 10-4mm, the time is accurate to 10-5s). The stress-strain curves were obtained for many times under the same deformation condition and have no great difference in the results, so the experiment is repeatable.
⑤ The displacement rate of the device is 0.1133 mm·s-1, 1.133 mm·s-1, 11.33 mm·s-1, 113.3 mm·s-1 respectively.
- Why has the difference in the strength of the sample decreased with the increase in the strain rate? As the strain rate increases, the most significant increase in strength is for the sample with the highest temperature. What is the reason? Why are these results not compared? This weakness can be seen in all parts of the article, including the abstract, results, discussion, and conclusion, which should be solved. How has the used model been verified?
Reply: Thank you for your valuable comments. The decrease of deformation rate provides enough time for dislocation movement and the formation of polygonal subgrain boundaries, resulting in the dynamic softening mechanisms, such as DRV and DRX. Higher deformation temperature provides greater driving force for dislocation movement. We add related discussion in the paper. (Lines 125-132,135-137)
- The results section is well organized and categorized. But some parts of it are reporting, and the discussion needs to be deepened. It is suggested to use the following sources. Co-precipitated Ni/Mn shell coated nano Cu-rich core structure: A phase-field study. Phase transformations in an ultralight BCC Mg alloy during anisothermal ageing. From classical thermodynamics to phase-field method. In the discussion section, only two sources have been used.
Reply: Thank you for the sources you provided. We have read them carefully and added them into the reference ([39,40]). And we have enriched our discussion and conclusion.

Round 2
Reviewer 1 Report
Accept
Few typo-mistakes should be corrected